# Hibernating bear serum hinders osteoclastogenesis *in-vitro*

**Alireza Nasoori[1], Yuko Okamatsu-Ogura[2], Michito Shimozuru[1], Mariko Sashika[1], Toshio Tsubota[1] ***

**1** Laboratory of Wildlife Biology and Medicine, Department of Environmental Veterinary Science, Graduate School of Veterinary Medicine, Hokkaido University, Sapporo, Japan, **2** Laboratory of Biochemistry, Department of Biomedical Sciences, Graduate School of Veterinary Medicine, Hokkaido University, Sapporo, Japan

\* tsubota@vetmed.hokudai.ac.jp

**Data Availability Statement:** All relevant data are within the manuscript.

**Funding:** This study was completely supported by a Grant-in-Aid from the Ministry of Education, Sports, Science, and Technology (MEXT) of Japan

## Abstract

Bears do not suffer from osteoporosis during hibernation, which is associated with long-term inactivity, lack of food intake, and cold exposure. However, the mechanisms involved in bone loss prevention have scarcely been elucidated in bears. We investigated the effect of serum from hibernating Japanese black bears (*Ursus thibetanus japonicus*) on differentiation of peripheral blood mononuclear cells (PBMCs) to osteoclasts (OCs). PBMCs collected from 3 bears were separately cultured with 10% serum of 4 active and 4 hibernating bears (each individual serum type was assessed separately by a bear PBMCs), and differentiation were induced by treatment with macrophage colony stimulating factor (M-CSF) and receptor activator of NF-kB ligand (RANKL). PBMCs that were cultured with the active bear serum containing medium (ABSM) differentiated to multi-nucleated OCs, and were positive for TRAP stain. However, cells supplemented with hibernating bear serum containing medium (HBSM) failed to form OCs, and showed significantly lower TRAP stain ($p < 0.001$). On the other hand, HBSM induced proliferation of adipose derived mesenchymal stem cells (ADSCs) similarly to ABSM ($p > 0.05$), indicating no difference on cell growth. It was revealed that osteoclastogenesis of PBMCs is hindered by HBSM, implying an underlying mechanism for the suppressed bone resorption during hibernation in bears. In addition, this study for the first time showed the formation of bears' OCs *in-vitro*.

## Introduction

Hibernation is a phenomenon affected by external stimuli such as cold season, short day length and food paucity, and internal factors such as biochemical regulations and circadian rhythm. Hibernating mammals have evolved with various types of physiological and behavioral responses, which enable them to go through harsh environmental conditions. Bears, in terms of body mass, are by far the largest hibernating mammalian species. Another distinctive feature is that bears do not intake food, urinate, nor defecate during hibernation. Interestingly,

(No. 17H03936: TT). There was no additional external funding received for this study.

**Competing interests:** The authors have declared that no competing interests exist.

hibernating bears' musculoskeletal system is not compromised by environmental and metabolic alterations throughout the torpor period [1–3].

Previous studies by the use of biochemical, histological, and imaging tests have clearly shown that bears do not suffer from osteoporosis despite long-term inactivity, lack of food intake and cold exposure during hibernation [4–10]. Similar conditions in humans and other mammals lead to osteopenia and osteoporosis, e.g. long-term inactivity and disuse such as long-term bed bound, and microgravity increase bone resorption to a rate higher than bone formation [11–15]. The imbalance in bone remodeling culminates in various degrees of bone loss.

There is growing evidence that the serum component can reflect the state of osteoporosis, and physical activity in humans [16–20]. The use of osteoporotic patients' serum in osteogenic cell culture can induce osteoporotic-like conditions *in-vitro*, such as increase in the expression of receptor activator of nuclear factor kappa-β ligand (RANKL) [21], a potent inducer of osteoclast (OC) differentiation, and decrease in the expression of osteogenic genes [22].

The use of bear serum for cell culture has been investigated in recent years. The *in-vitro* experiments have indicated that the use of active and hibernating bear serum for cell culture can induce active-like and hibernation-like cell responses, respectively [23–25]. For example, the use of hibernating bear serum inhibits proteolysis, and increases protein content in muscle cells [25].

As mentioned above, it is almost unequivocal that bears do not develop osteoporosis due to hibernation. However, the mechanisms and features that enable bears to stay safe from bone loss have largely remained unknown. Here, we present *in vitro* evidence that serum of hibernating bears suppresses osteoclastogenesis of bear peripheral blood mononuclear cells (PBMCs). PBMCs, akin to bone marrow monocytes, are considered as progenitor cells for osteoclasts. Also, here for the first time we report the formation of bear OCs *in-vitro*.

## Materials and methods

### Sampling from animals

We used captive adult Japanese black bears (*Ursus thibetanus japonicus*) at Ani Mataginosato Bear Park (Kuma Kuma) located in Akita Prefecture, Japan (N 39.915398 E140.536294), in active and hibernation periods. Bear information and sample types are presented in Table 1. The bears are routinely fed by crushed corn, acorn, chestnut, and a kind of local Japanese butterbur. During hibernation period, the bears do not have access to food, while they have access to water ad libitum. Bears hibernation was characterized by lack of movement and food intake. Hibernation in these captive bears reportedly begins in December and proceeds to around

**Table 1. Bear information, sample type, and collection season.**

| Bears ID | A | B | C | K | D | E | F | G | C | D | H | I |
|---|---|---|---|---|---|---|---|---|---|---|---|---|
| Type of sample | PBMCs | PBMCs | PBMCs | ADSCs | Serum | Serum | Serum | Serum | Serum | Serum | Serum | Serum |
| Season | May | May | July | July | May | May | May | May | January | January | January | January |
| Age (year) | 7 | 7 | 19 | 23 | 18 | 12 | 26 | 16 | 19 | 18 | 18 | 17 |
| Body mass* (kg) | 86 | 90.5 | 143 | 157 | 145.5 | 153 | 107 | 78 | 157.5 | 166 | 135.5 | 201 |

All samples were collected from male bears, except bear G, a female bear. May and July samples are for active phase. January samples are for hibernation phase. The average of active and hibernating bears' age (for serum samples) is 18.00 ± 2.94 and 18.00 ± 0.40, respectively. The difference between the groups' age is not significant; *p*-value = 1.00. Data were assessed by two-tailed Student's *t*-test.

*Body mass at sampling season.

early April, when bears resume activity and foraging [26,27]. All procedures and animal care were conducted in accordance with the Guideline of the Animal Care and Use of Hokkaido University and were approved by the Animal Care and Use Committee of Hokkaido University (Permit Number: 18–0179). To collect samples, bears were anesthetized with 40 μg/kg medetomidine hydrochloride (Domitor; Zenoaq, Fukushima, Japan) and 3.0 mg/kg zolazepam hydrochloride and tiletamine hydrochloride cocktail (Zoletil; Virbac, Carros, France) intramuscularly. Blood was collected from the jugular vein into plain tubes, and EDTA containing tubes for serum collection and peripheral blood mononuclear cells (PBMCs) isolation, respectively. Serum was isolated from plain tubes and preserved at -80˚C. EDTA containing tubes were delivered at 4˚C to the lab for PBMCs isolation, as explained below. Adipose tissue was biopsied from the subcutaneous inguinal and sternal areas, put in Dulbecco's Modified Eagle's Medium (DMEM, FUJIFILM Wako Pure Chemical Corporation, Japan), and delivered at room temperature to the lab for adipose derived mesenchymal stems cells (ADSCs) isolation, as explained below.

## PBMCs isolation

The blood samples were obtained from 3 adult male bears A, B, and C in active season (May-July). In the lab, the blood (collected in EDTA containing tubes) collected from each bear was separately diluted with phosphate-buffered saline (PBS) to the ratio 50:50. Blood-PBS mixture was slowly added onto Ficoll-Paque PLUS (GE Healthcare Life Sciences, Sweden) in falcon tubes based on the instruction of the product. The tubes were centrifuged at $400 \times g$, for 40 minutes at 20˚C. The buffy coat was collected and transferred to new falcon tubes, and were centrifuged twice with PBS at $400 \times g$ for 15 and 10 minutes at 20˚C. Finally, the PBMCs from each bear were separately collected in Eppendorf tubes, and preserved in CELLBANKER 2 (AMS Biotechnology Ltd., UK) at -80˚C.

## ADSCs isolation

Adipose tissue was obtained from a mature male black bear (bear K) in active season. The tissues were digested by 2 mg/ml collagenase (FUJIFILM Wako Pure Chemical Corporation, Japan) in DMEM containing 2% fatty acid-free bovine serum albumin (BSA) (FUJIFILM Wako Pure Chemical Corporation) at 37 ºC for 2 h with shaking at 90 cycles/min. The suspension was filtered through 200-μm nylon filter and centrifuged at $200 \times g$ for 5 min at room temperature. The pellet was filtered through 25-μm nylon filter, and then centrifuged at $200 \times g$ for 5 min at room temperature. The pellet was collected as ADSCs, and preserved in CELLBANKER 2 (AMS Biotechnology Ltd., UK) (Takara Bio Inc., Japan) at -80˚C. This method is based on our previously established protocol for ADSCs isolation from adipose tissue [28].

## Cell culture

PBMCs were cultured at a concentration of 3 to $5 \times 10^6$ cells per 35 mm collagen-coated dish, in a-MEM (Minimum Essential Medium Eagle, Sigma-Aldrich, UK.) supplemented with 10% bear serum, 100 units/ml penicillin-streptomycin (FUJIFILM Wako Pure Chemical Corporation), 50 μg/ml gentamicin (FUJIFILM Wako Pure Chemical Corporation), 2.5 μg/ml amphotericin B (FUJIFILM Wako Pure Chemical Corporation) and 2 mM L-Glutamine (Nacalai Tesque Inc., Japan), and incubated at 37˚C with 5% $CO_2$. Protocol for the OC differentiation is summarized in Fig 1. The day after seeding (day 2), the supernatant was discarded, and cells either i) at a concentration of 0.2 to $0.5 \times 10^5$ cells per well were seeded on 96-well plate or ii) at the same dish remained to the end of experiment, were cultured with the same medium/

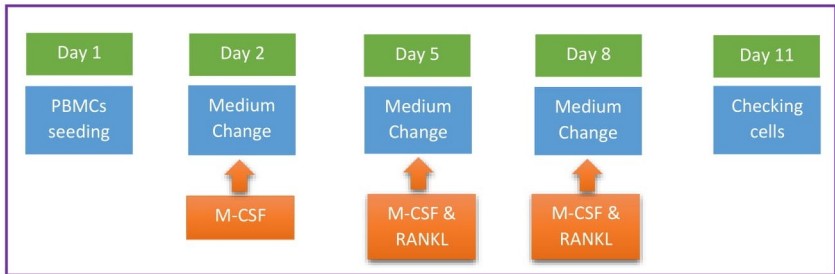

**Fig 1. The procedure of PBMCs culture from day 1 to 11 to form OCs.**

serum type with addition of 20 ng/ml macrophage colony stimulating factor (M-CSF, Gibco, USA.) for three days. On day 5 and 8, the medium was changed to new medium containing 20 ng/ml M-CSF and 50 ng/ml RANKL (Sigma-Aldrich, USA.). On day 11, cells were finally washed with PBS, fixed with 10% phosphate-buffered formalin, washed with distilled water, and stained with TRAP Stain (TRAP staining kit, Cosmobio Co., LTD., Japan) to examine the presence of tartrate-resistant acid phosphatase (TRAP) by a light microscope. The stained area in each dish was quantified by the use of NIH Image J Software.

PBMCs of bears A, B, and C were separately cultured with active bear serum containing medium (ABSM) prepared separately from serum of bears D, E, F, and G, and hibernating bear serum containing medium (HBSM) prepared separately from serum of bears C, D, H, and I (Table 1). PBMCs of bear A and B were twice evaluated for the mentioned procedure on dish and plate. All groups were treated with MCSF and RANKL equally and similarly.

ADSCs from one bear (bear K) were cultured on 35 mm collagen-coated dish (cell concentration shown in the results) with 10% ABSM or 10% HBSM, each from 4 different individuals, as mentioned for PBMCs. ADSCs were twice cultured; after 6 days (medium changed at day 3), cells were detached by trypsin/EDTA solution, counted on hemocytometer, re-cultured (subculture), and after 4 days, detached and recounted. At each passage, ADSCs were cultured separately with each of four types of ABSM and HBMS.

## Statistical analysis

Values are expressed as mean ± standard error (SE). The data were analyzed by the use of analysis of two-tailed Student's *t*-test or Analysis of Variance (ANOVA) together with a post-hoc Tukey HSD (by the use of IBM SPSS Statistics, Version 23), and significant *p*-value was considered equal or less than 0.05.

## Results

### PBMCs culture and TRAP staining

PBMCs from bears A, B, and C (Table 1) were cultured with either ABSM or HBSM, and induced differentiation to OCs by treatment with M-CSF and RANKL (Fig 1). On day11, PBMCs cultured with ABSM showed multi-nucleated morphology, a typical feature of OCs. These multi-nucleated cells were positively stained for TRAP activity, indicating differentiation to active OCs (Fig 2A and 2B). By contrast, PBMCs cultured with HBSM failed to differentiate to multi-nucleated cells (Figs 1C and 2D). TRAP-stained area in ABSM group was 21.28 ± 3.89% of total area, being significantly (*p*-value < 0.001) higher than that in HBSM group (1.73 ± 0.43%) (Fig 3).

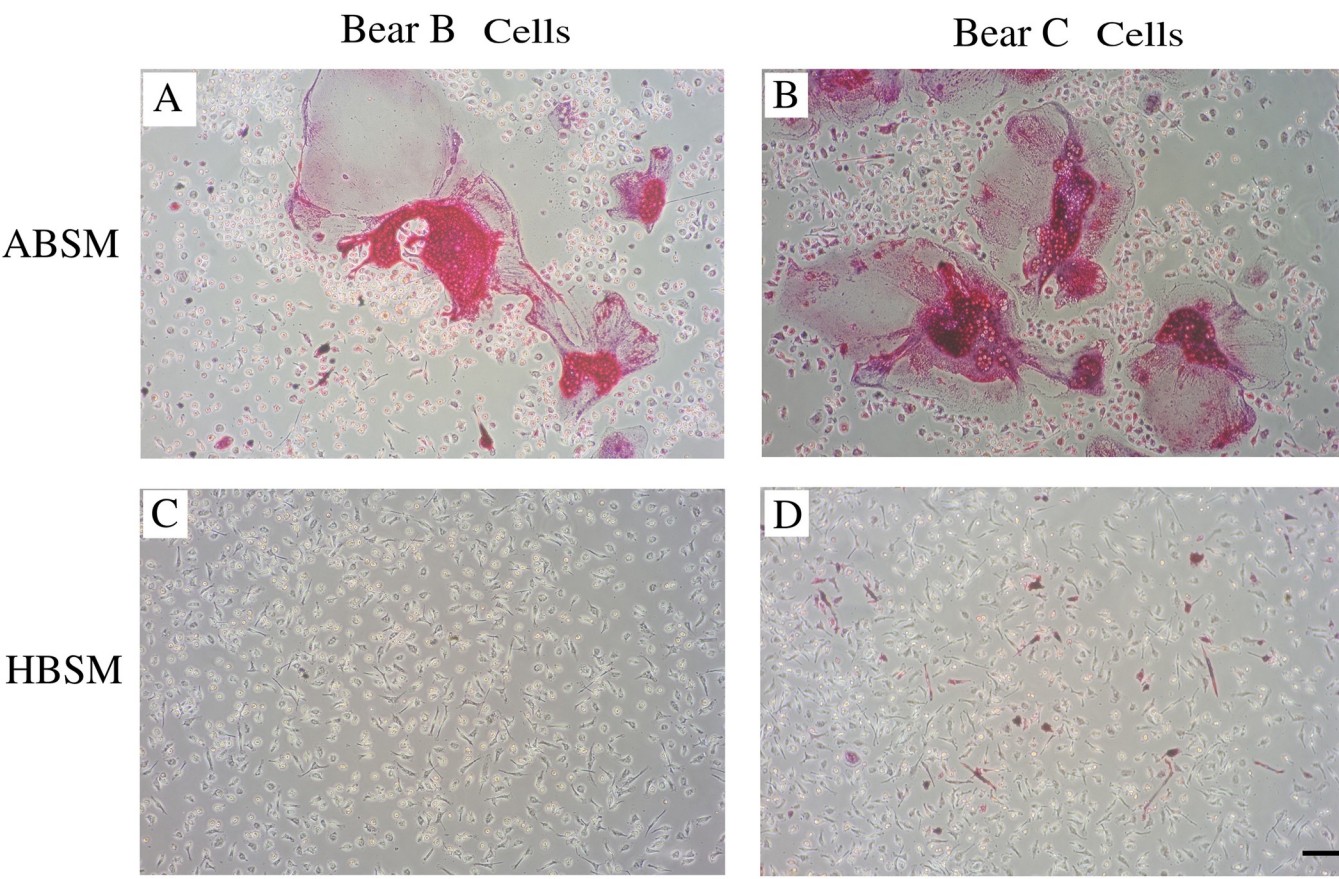

**Fig 2. TRAP stained samples at day 11.** (A-B) cultured cells of bears B and C with active bear serum containing medium (ABSM); multi-nucleated giant osteoclasts which are TRAP stain positive. (C-D) cultured cells of bears B and C with hibernating bear serum containing medium (HBSM); non-differentiated cells, which are poorly stained with TRAP stain. Scale bar is equal to 500 μm.

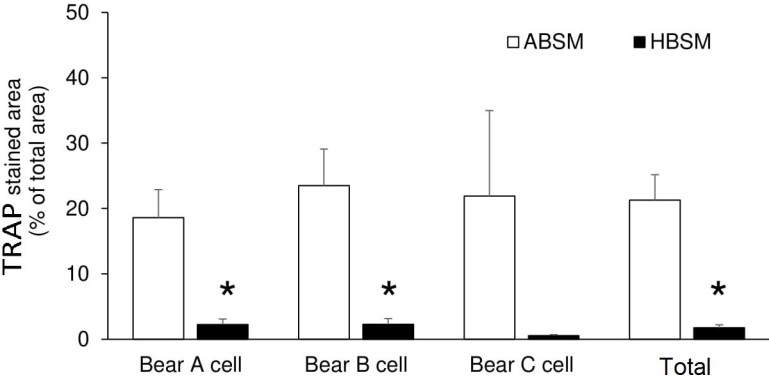

**Fig 3. The percentage of TRAP stained areas in osteoclast cultures.** Bear A cultured with active bear serum containing medium (ABSM) (n = 4), and hibernating bear serum containing medium (HBSM) (n = 4). * $p$-value = 0.040. Bear B cultured with ABSM (n = 4), and HBSM (n = 4). * $p$-value = 0.028. Bear C cultured with ABSM (n = 3), and HBSM (n = 3). $p$-value, = 0.242. Total of three bears: ABSM (n = 11), and HBSM (n = 11). * $p$-value = 0.0005. Data were assessed by two-tailed Student's $t$-test. All values represent means ± SE.

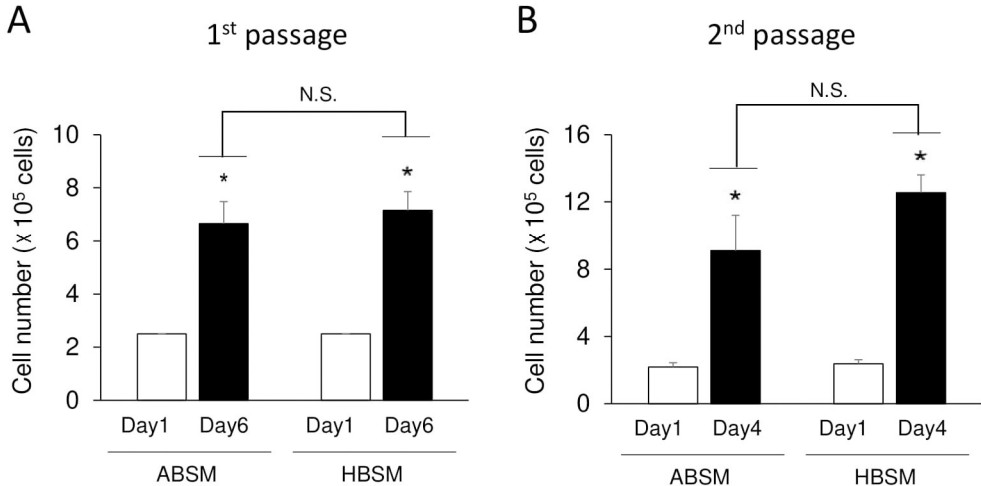

**Fig 4. The cell number of Adipose Derived Stem Cells (ADSCs) of bear K cultured with active bear serum (separately with each serum of bear D, E, F, and G) containing medium (ABSM) and hibernating bear serum (separately with each serum of bear C, D, H, I) containing medium (HBSM) for the first and second passage.** (A) First passage; there was a significant difference between groups as determined by one-way ANOVA (F(3–12) = 21.80, *p*-value <0.001). Post-hoc comparisons using Tukey HSD test indicated that there was a significant difference between day 1 and 6 in ABSM group *p*-value = 0.001 and in HBSM group * *p*-value < 0.001. The difference between ABSM and HBSM at day 6 was not significant (N.S); *p*-value = 0.914. (B) Second passage; there was a significant difference between groups as determined by one-way ANOVA (F(3–12) = 18.59, *p*-value <0.001). Post-hoc comparisons using Tukey HSD test indicated that there was a significant difference between day 1 and 4 in ABSM group * *p*-value = 0.007, and in HBSM group * *p*-value < 0.001. The difference between ABSM and HBSM at day 4 was not significant (N.S); *p*-value = 0.224. Each column shows the average of total cell numbers of 4 separate cultures based on 4 types of ABSM and 4 types of HBSM. All values represent means ± SE.

## ADSCs culture and cell number count

ADSCs were cultured with either ABSM or HBSM and effect on cell proliferation was examined. At the first passage, 6-day culture showed significant (*p*-value < 0.05) increase in cell number (Fig 4A) and the growth rate showed no significant difference between ADSCs cultured with ABSM (2.7-fold) and HBSM (2.9-fold) (Fig 4A). At the second passage, cell number significantly (*p*-value < 0.05) increased in both ABSM and HBSM (Fig 4B). Growth rates of ABSM and HBSM groups were similar in 4-day culture at the second passage (Fig 4B). ADSCs in both groups turned to spindle-shaped cells at day 6 (first culture), and day 4 (subculture/ second culture) (Fig 5).

## Discussion

PBMCs and bone marrow derived monocytes/macrophages (BMM) are ubiquitously used for *in-vitro* osteoclastogenesis. Previous investigations have presented *in-vitro* OC formation from different mammalian species such as humans [29], primates [30], rodents [31], dogs [32], cats [33,34], rabbits [35], horses [36], and pigs [37]. To the best of our knowledge, the current work is the first to report bears' OC formation *in-vitro*. In this study we used bears' PBMCs for OC formation *in-vitro*. In order to obtain PBMCs, only blood should be collected, which is a low risk, less invasive and inexpensive method in comparison with bone marrow collection method for isolation of BMM, particularly for wild animals such as bears.

In *in-vitro* culture, under the influence of M-CSF and RANKL treatment, PBMCs differentiate to macrophage/pre-OCs, which is followed by cell fusion. Mature and functional OCs are characterized by the presence of multiple nuclei in a very large unified cell (multi-nucleated giant cells) which produce TRAP (TRAP stain positive) (Fig 2A and 2B) [38–42].

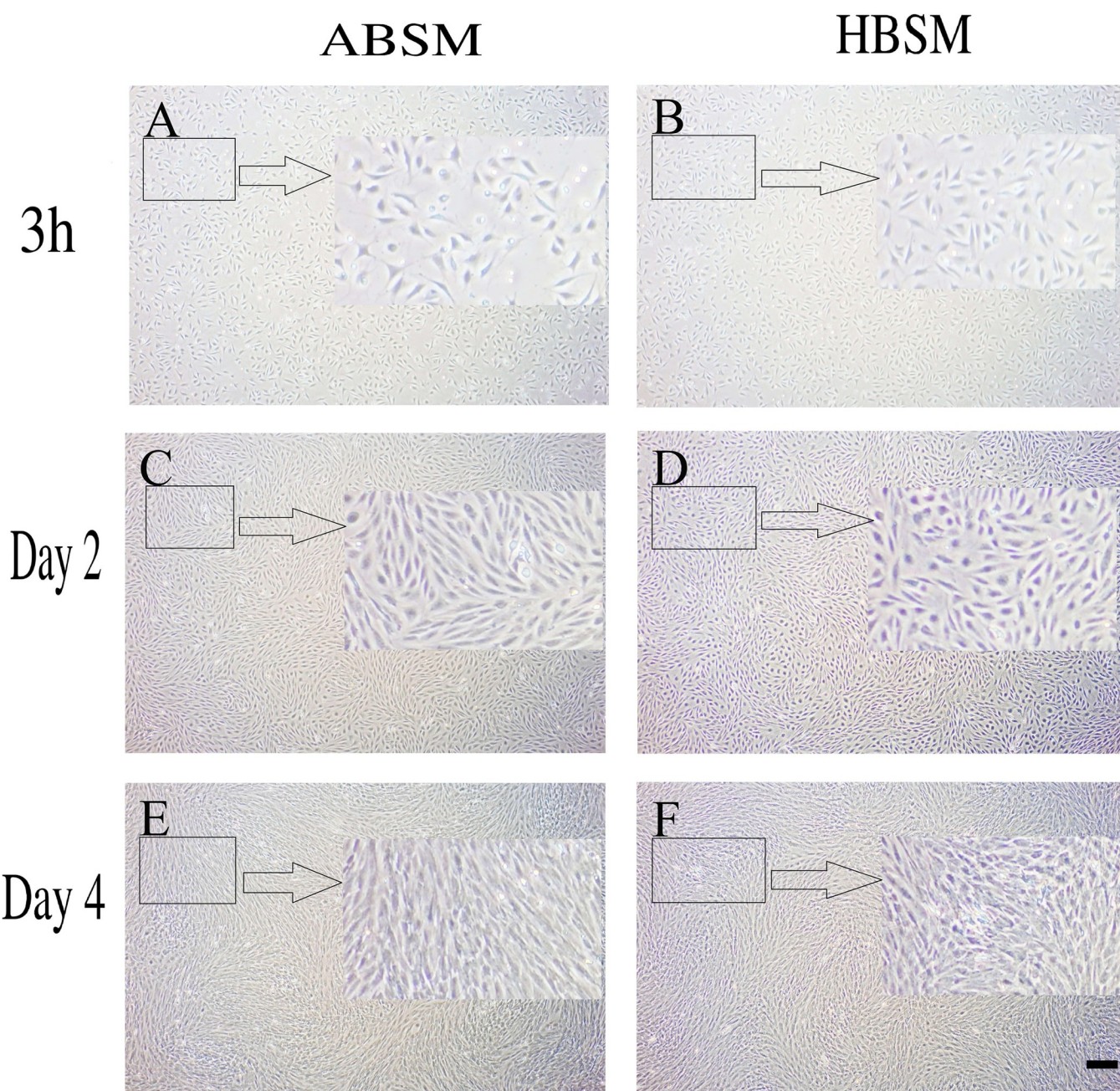

**Fig 5. Second passage of Adipose Derived Stem Cells (ADSCs) cultured with active bear serum containing medium (ABSM) and hibernating bear serum containing medium (HBSM) in time intervals: 3 hours, day 2 and day 4 post seeding.** This indicates a similar trend in cell growth for both ABSM and HBMS. Scale bar is equal to 500 μm. Arrows show 100% zoomed areas in the square.

Here we demonstrated that bears' PBMCs that were cultured with ABSM could differentiate to OCs and were significantly TRAP positive. However, PBMCs that were cultured with HBSM failed to differentiate to OCs, and were almost negative to TRAP staining (Figs 2 and 3). This is consistent with a previous study demonstrating that denning (hibernating) bears' serum derived components (isolates) inhibited chicken BMM from OCs formation [43]. Biochemical experiments indicated that the genes which contribute to OCs formation and

differentiation (such as *Ostf1*, *Rab9a*, and *c-Fos*) are underexpressed in hibernating bears [9]. Also, the level of serum TRAP (a bone resorption biomarker) is significantly lower in hibernation period than before and after hibernation in bears [10]. In addition, histological and imaging tests revealed that trabecular and compact bone mass of the front and hind limbs are not compromised in hibernating bears. The bone mass preservation has been suggested to be related to a reduced, yet balanced, bone turnover during hibernation [4,5,6,8].

In the present work, we observed that HBSM group could not differentiate into OCs. Both ABSM and HBSM groups were treated similarly; the only variable was the type of serum which was either hibernation or normal type. This could, in part, made us speculate that "serum factors that modulate cell differentiation are compromised during winter dormancy due to lack of food intake". In other words, we speculated that hibernation phase serum is not "rich enough" for cell culture. Hence, in another experiment, we tested both ABSM and HBSM on bears' adipose-derived stem cell culture (ADSCs). Interestingly, treatment of a bear ADSCs with both serum types of the same 8 bears (4 ABSM and 4 HBSM) indicated significant cell growth, and formation of spindle-like mesenchymal cell (Figs 4 and 5). We also ruled out the effect of culture dish where we used 35 mm collagen-coated dish for both PBMCs and ADSCs culture. These findings indicate that HBSM is "not poor" for ADSCs culture; however, HBSM (unlike ABSM) fails to promote PBMCs culture.

Previous investigations on bear adipocyte culture by active and hibernating bear serum represented metabolic profiles similar to *in-vivo* [24]. In addition, bears' ADSCs have shown optimal differentiation to different cell lineages such as chondroblasts and osteoblasts [44,45]. Human osteoblasts that were cultured with bear serum proliferate well [46,47]. Furthermore, the osteoblasts show similar response to hibernation and active bear serum [46,47]. Altogether, it can be deduced that while hibernating phase serum hinders OC formation, it retains the proliferative and differentiating potential for ADSCs and osteoblasts. This may, in part, participate in hibernating bears' bone mass maintenance.

Still, it remains questionable how hibernating phase serum can hinder osteoclastogenesis, and which serum factors or mechanisms are involved in this phase. Studies on hibernating bears have denoted that there are various serum factors that play roles in bone metabolism. For instance, *i*) Parathyroid hormone (PTH) is thought to have major anabolic effects on bone metabolism in American black bears (*Ursus americanus*) [2,48]. Gray et al. [49] reported that treatment of dystrophin-deficient mice with black bear's PTH decreased osteoclastic surface and increased osteoblastic surface on the bone, leading to significantly higher hind limb bone density than control group [49]. *ii*) It has been reported that melatonin concentration during hibernation is about 7.5 times higher than summer active phase in brown bears (*Ursus arctos*) [50]. Melatonin has indicated inhibitory effects on osteoclastogenesis *in-vitro*. This was partially attributed to the increased osteoprotegerin: RANKL ratio, by inhibiting RANKL expression in osteoblasts [51–53]. *iii*) It is also possible that serum factors affect bone metabolism indirectly through the modulation of circadian rhythm. Jansen et al., [23] showed when bear's fibroblasts were cultured with active and hibernation phase serum containing medium, cell molecular rhythm was akin to those of active and hibernation phases, respectively. Circadian rhythm can govern OC activity and formation through different pathways, such as: 1) Regulation of aryl hydrocarbon receptor nuclear translocator-like (Bmal1) in OC [54], 2) Modulation of Bmal1 in osteoblasts which leads to alteration in RANKL expression [55], and 3) β-adrenergic and glucocorticoid signaling in OC [56,57]. *iv*) Black bears' serum immune related factors are altered during hibernation, some of which are thought to affect bone metabolism, e.g. α2-HS-glycoprotein (AHSG), which is an osteogenic inhibitor, is downregulated in winter dormancy [58].

The present study for the first time revealed the influence of HBSM on osteoclastogenesis. There are more serum factors than stated above, which are likely to be involved in hampering OC formation/activity, and general bone metabolism [2,48]. These demand very large-scale tests to examine the efficacy of the serum factors on OC progenitors. In our preliminary tests, we examined the heat inactivation of bear serum on PBMCs culture. These tests failed to show any differentiation in PBMCs culture either with heat-inactivated ABSM or heat-inactivated HBSM. However, when heat-inactivated ABSM and HBSM were applied to ADSCs, the cell growth was comparable with normal (unheated) ABSM and HBSM. It has been suggested that the failure in cell culture with heat inactivation of serum might be due to some cell lineages [59]. It is noteworthy that bone remodeling includes bone resorption and bone formation, undertaken by osteoclasts and osteoblasts, respectively. Former studies suggested that both bone formation and resorption are equally suppressed in hibernating bears; an equilibrium which sustain bone remodeling at a low rate in order to save energy for a long-term torpor [4,8–10]. The present study shed light on OCs, indicating that bears PBMCs under the influence of HBSM fail to form OC in *in-vitro* condition. Osteoclastogenesis markers such as β3 integrin, cathepsin K, TRAP, and calcitonin are required to be explored regarding bears PBMCs differentiation in the future studies. It must be noted that, PBMCs are not the only source for OC formation *in-vivo*, i.e. bone marrow monocytes, and bone macrophages also account for osteoclastogenesis and bone resorption [60–63]. Therefore, the effect of HBSM on other OC progenitors remains to be explored. Moreover, further investigations are required to demonstrate biochemical alterations that regulate osteoclastogenesis during winter dormancy.

The evolutionary trends and physiological attributes that enable hibernating mammals such as bears [4,5], ground squirrels [64], marmots [65], and woodchucks [66] to preserve bone mass deserve to be explored. In comparison with other hibernating mammals, bears have a large body size, near-to-normal body temperature during hibernation, and a drastic pre-hibernation fat storage [26,27]; such peculiar differences may bring about bone preservation mechanisms that other hibernating mammals do not possess. McGee-Lawrence (2011) stated that although hibernating ground squirrels preserve macrostructural cortical bone geometry and strength, they undergo trabecular, and cortical bone loss on a microstructural scale during hibernation, which is different from hibernating bears bone condition [64]. Such differences might be due to species-specific osteogenic alterations, which have also been reported in other mammals. For instance, a transient physiological osteoporosis due to a rapid antler osteogenesis may occur in cervids, which is a distinct feature in comparison with other artiodactyls [67]. Hence, comprehensive and comparative studies on mammalian physiological bone preserving/ or osteoporotic mechanisms are needed to provide insights into mammalian bone biology.

## Acknowledgments

We wish to thank the staff, especially Mr. Akihiro Satoh and Dr. Takeshi Komatsu, at the Ani Mataginosato Bear Park for their generous support.

## Author Contributions

**Conceptualization:** Alireza Nasoori.

**Data curation:** Alireza Nasoori, Yuko Okamatsu-Ogura, Toshio Tsubota.

**Formal analysis:** Alireza Nasoori, Yuko Okamatsu-Ogura.

**Funding acquisition:** Toshio Tsubota.

**Investigation:** Alireza Nasoori, Yuko Okamatsu-Ogura, Toshio Tsubota.

**Methodology:** Alireza Nasoori, Yuko Okamatsu-Ogura, Michito Shimozuru.

**Project administration:** Toshio Tsubota.

**Resources:** Toshio Tsubota.

**Supervision:** Yuko Okamatsu-Ogura, Michito Shimozuru, Mariko Sashika, Toshio Tsubota.

**Validation:** Alireza Nasoori, Yuko Okamatsu-Ogura, Toshio Tsubota.

**Visualization:** Yuko Okamatsu-Ogura.

**Writing – original draft:** Alireza Nasoori.

**Writing – review & editing:** Yuko Okamatsu-Ogura, Michito Shimozuru, Mariko Sashika, Toshio Tsubota.

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
