## [Decision Letter · Decision Letter 0]

11 Jun 2020

PONE-D-20-13988

Hibernating bear serum hinders osteoclastogenesis in-vitro

PLOS ONE

Dear Dr. Tsubota,

Thank you for submitting your manuscript to PLOS ONE. After careful consideration, we feel that it has merit but does not fully meet PLOS ONE’s publication criteria as it currently stands. Therefore, we invite you to submit a revised version of the manuscript that addresses the points raised during the review process.

We look forward to receiving your revised manuscript.

Kind regards,

Xing-Ming Shi, Ph.D

Academic Editor

PLOS ONE

Journal Requirements:

"This study was supported in part by a Grant-in-Aid from the Ministry of Education, Sports, Science, and Technology (MEXT) of Japan (No. 17H03936: TT). MEXT also provided the scholarship to AN.".

i) Please provide an amended statement that declares *all* the funding or sources of support (whether external or internal to your organization) received during this study, as detailed online in our guide for authors at http://journals.plos.org/plosone/s/submit-now.  Please also include the statement “There was no additional external funding received for this study.” in your updated Funding Statement.

ii) Please include your amended Funding Statement within your cover letter. We will change the online submission form on your behalf.

Reviewers' comments:

Reviewer's Responses to Questions

**Comments to the Author**

1. Is the manuscript technically sound, and do the data support the conclusions?

Reviewer #1: Yes

Reviewer #2: Partly

2. Has the statistical analysis been performed appropriately and rigorously? 

Reviewer #1: Yes

Reviewer #2: Yes

3. Have the authors made all data underlying the findings in their manuscript fully available?

Reviewer #1: Yes

Reviewer #2: Yes

4. Is the manuscript presented in an intelligible fashion and written in standard English?

Reviewer #1: Yes

Reviewer #2: Yes

5. Review Comments to the Author

Reviewer #1: The authors have investigated the effects of serum isolated from active and hibernating bears for osteoclastogenesis in vitro and observed that hibernating bear serum failed to promote osteoclastogenesis from PBMCs cultured with M-CSF and RANKL, although active bear serum promote that. In addition, the osteoclast culture from bears’ PBMCs is significant. The manuscript is written in a clear and concise style. However, some concerns were raised.

Major issues:

The authors evaluated osteoclastogenesis only by areas of TRAP-positive cells in Figure 2 and 3. However, the reviewer feels that it is a little tenuous to determine the conclusion drawn from this data. Osteoclasts are multinucleated cells of differentiation and fusion of monocyte-macrophage precursors. The magnitude of bone resorption by osteoclasts is regulated by their number and the resorptive capacity of the individual cells. Therefore, measurements of the number of TRAP-positive multinucleated cells and the expression of osteoclastogenic markers such as β3 integrin, cathepsin K, TRAP, and calcitonin receptor could be more definitive evidence to support the data in Figure 2 and 3.

Minor issues:

1. Did the authors separately use the PBMCs and serum from each bear? Please provide the detail sample numbers and replicates assigned for each experiment in the text of manuscript.

2. The authors should provide cell number or concentration seeded for osteoclast culture.

3. About figure 4, panels C-F, described in the text, are not found in the figure. The authors should check them.

Reviewer #2: This is an interesting study that is consistent with osteoclastogenesis in peripheral blood mononuclear cells being influenced by serum factors in hibernating and active bears. This builds on some previous research with chicken osteoclastogenesis being affected by bear serum and the observation that bears do not suffer significant bone loss during hibernation. There are some troubling experimental design flaws in the paper that should be addressed before publication.

Introduction

Line 72: ground squirrels do not eat during hibernation and rarely urinate or defecate.

Line 80: it would be good to compare and contrast bears to what is known about bone loss in other hibernators such as ground squirrels. For example: McGee-Lawrence ME, Stoll DM, Mantila ER, Fahrner BK, Carey HV, Donahue SW. Thirteen-lined ground squirrels (Ictidomys tridecemlineatus) show microstructural bone loss during hibernation but preserve bone macrostructural geometry and strength. J Exp Biol 214: 1240–1247, 2011.

Line 82: change human to humans

Methods

Line 123: add “have” in front of “access” in both occurrences

Line 129: was blood collection done similarly for both hibernating and alert animals? Some anesthetics don’t work as well on hibernating animals due to decreased circulation and metabolism

Line 131: if blood was collected in EDTA this would produce plasma. Was clotting initiated to form serum? Or was plasma actually used in the experiments.

Line 174: Change “Next day of” to “The day after”

For the experimental design (table 1 and lines 188-194)

I have a few concerns with the experimental design.

1. While an n=3 for PBMCs is fairly low an n=1 for ADSC is unacceptably low. It also appears that individual PBMC cell preps were incubated with serum from individual animals. Were any experiments performed where a single cell prep was incubated with each of the different serum samples to ensure there were not differences between serum samples. Or a pooled serum sample could be used in common with all three cell preps. As the experiment is designed it is difficult to tell if any variation is due to the cell prep or the serum.

2. The difference in growth between active (ABSM) and hibernating (HBSM) sera is interesting and suggests that ABSM contains an activator or HBSM contains an inhibitor of PBMC differentiation. A classical experimental method to test this is to either heat inactivate each serum or do a mixing experiment. For example if you heat inactivate ABSM and growth no longer occurs it could contain an activator, while if you heat inactivate HBSM and growth does occur then it contained an inhibitor. Similarly, if HBSM contains an inhibitor, adding it to ABSM could decrease growth. There are multiple ways these experiments may not work (heat stable factors, etc.) but they are fairly simple and broad experiments that would allow the authors to draw a more substantial conclusion about their results.

Results: The images are not of great quality, but if they meet the standards of the journal they should be sufficient. My major concern with the results are addressed above regarding the sample size, especially for ADSC, and the mixing and heat inactivation controls.

Discussion: A quick summary of the use of PBMCs in in vitro osteoclastogenesis should be added to the introduction. My first thought when reading the introduction is whether this model was physiologically relevant, i.e. how many osteoclasts arise from PBMCs vs. bone marrow. If this is the standard of the field it would be good to mention this up front.

The mixing and heat inactivation experiments are even more important to perform given the mention on line 296 that hibernating serum inhibits chicken OCs formation.

The authors come very close to concluding that HBSM might contain an OC inhibitor in lines 304-313. The experiments described above would allow them to test this hypothesis and make this conclusion directly.

The rest of the discussion speculates on possible factors that may be lacking in hibernating serum. I think the authors would be advised to first determine if an activator is lacking or an inhibitor is present before testing individual potential lead compounds.

6. PLOS authors have the option to publish the peer review history of their article (what does this mean?). If published, this will include your full peer review and any attached files.

Reviewer #1: No

Reviewer #2: No

---

## [Author Response · Author response to Decision Letter 0]

26 Jul 2020

Dear Reviewers,

I and co-authors would like to thank you for the constructive comments. The changes in the text were highlighted in yellow. The added references (reference number) in the text and reference list were highlighted in red. All the comments are responded as follows:

Regards,

Corresponding author

Reviewer #1: 

The authors have investigated the effects of serum isolated from active and hibernating bears for osteoclastogenesis in vitro and observed that hibernating bear serum failed to promote osteoclastogenesis from PBMCs cultured with M-CSF and RANKL, although active bear serum promote that. In addition, the osteoclast culture from bears’ PBMCs is significant. The manuscript is written in a clear and concise style. However, some concerns were raised.

► Thank you very much for reviewing the present paper.

Major issues:

1# The authors evaluated osteoclastogenesis only by areas of TRAP-positive cells in Figure 2 and 3. However, the reviewer feels that it is a little tenuous to determine the conclusion drawn from this data. Osteoclasts are multinucleated cells of differentiation and fusion of monocyte- macrophage precursors. The magnitude of bone resorption by osteoclasts is regulated by their number and the resorptive capacity of the individual cells. Therefore, measurements of the number of TRAP-positive multinucleated cells and the expression of osteoclastogenic markers such as β3 integrin, cathepsin K, TRAP, and calcitonin receptor could be more definitive evidence to support the data in Figure 2 and 3.

► In the present study, we considered two important (globally accepted) osteoclast features: multinucleated cell formation, and TRAP stainability. we asserted that the use of active phase serum (ABSM) leads to formation of multinucleated cells, while hibernation phase serum (HBSM) fails to form multinucleated cells, Line 54-57, 227-230, 284-286, i.e. presence of multinucleated cells with ABSM, and absence of multinucleated cells with HBSM. After TRAP stain, in order to make a quantitative comparison, the rate of stain absorption was compared between ABSM and HBSM cultured cells. 

Figure 2 shows the presence and absence of multinucleated cells (morphology). Figure 3 shows the functionality of cells by TRAP stain. Although TRAP stain is not the only criterion to show functionality, TRAP stain is a critical parameter which indicates that cells are active, and have the potential for resorption (Sources: https://doi.org/10.1007/BF02411252;
https://doi.org/10.1007/BF01623458;
https://doi.org/10.1080/08916930701694667). 

In agreement with your statement, specific makers such as cathepsin K, TRAP, etc. can reveal the pathways which ABSM is effective on osteoclastogensis. This is a very important part of our future study regarding evaluation of underlying mechanisms which are involved in “lack of osteoclastogenesis with HBSM”. These have been added to the revised manuscript within Discussion, Line 351, 352. We will definitely consider your comments for our subsequent studies to explore the osteoclastogenesis mechanisms in bears’ PBMCs by evaluation of bears’ specific mRNA. However, the scope of the present work is to compare the effect of HBSM and ABSM on bears PBMCs. In addition we evaluated the serum effects on adipose derived stem cell response to clarify the difference in cell response (if any) to HBSM and ABSM. 

Minor issues:

1. Did the authors separately use the PBMCs and serum from each bear? Please provide the detail sample numbers and replicates assigned for each experiment in the text of manuscript.

► Yes, separately, and with replicates. Corrections have been made in Line 50-52, 153, 158, 191-194, 200, 201, Caption of Fig. 4. 

Each cell (A, B, and C) were separately tested. Each cell was separately tested with each of 4 types of ABSM and HSBM. For example: 

PBMCs of Bear A was cultured with ABSM of Bear D

PBMCs of Bear A was cultured with ABSM of Bear E

PBMCs of Bear A was cultured with ABSM of Bear F 

PBMCs of Bear A was cultured with ABSM of Bear G

PBMCs of Bear A was cultured with HBSM of Bear C

PBMCs of Bear A was cultured with HBSM of Bear D

PBMCs of Bear A was cultured with HBSM of Bear H

PBMCs of Bear A was cultured with HBSM of Bear I

Similar procedure were used for PBMCs of Bears B and C. 

2. The authors should provide cell number or concentration seeded for osteoclast culture.

► Done. Line 172, 178, 179, 196, 197.

3. About figure 4, panels C-F, described in the text, are not found in the figure. The authors should check them.

► Corrected. Line 242-246. 

Reviewer #2: 

This is an interesting study that is consistent with osteoclastogenesis in peripheral blood mononuclear cells being influenced by serum factors in hibernating and active bears. This builds on some previous research with chicken osteoclastogenesis being affected by bear serum and the observation that bears do not suffer significant bone loss during hibernation. There are some troubling experimental design flaws in the paper that should be addressed before publication.

► Thank you very much for reviewing the present paper.

Introduction

1# Line 72: ground squirrels do not eat during hibernation and rarely urinate or defecate.

► Yes, Good point. This paragraph is about the bears features. The statements about ground squirrels come in Discussion, Line 358-371. 

One more thing is that, according to McGee-Lawrence et al., (2011) https://doi.org/10.1242/jeb053520, Page 1245:

“..other hibernators, such as ground squirrels, interrupt torpor bouts with periodic arousals to euthermia, during which they excrete calcium-containing waste. This provides a potential avenue for calcium and other products of bone catabolism to be excreted from the body, as occurs during other disuse situations such as human bedrest.” This statement contradicts the physiologic response of hibernating bears. 

2# Line 80: it would be good to compare and contrast bears to what is known about bone loss in other hibernators such as ground squirrels. For example: McGee-Lawrence ME, Stoll DM, Mantila ER, Fahrner BK, Carey HV, Donahue SW. Thirteen-lined ground squirrels (Ictidomys tridecemlineatus) show microstructural bone loss during hibernation but preserve bone macrostructural geometry and strength. J Exp Biol 214: 1240–1247, 2011.

► The statements about ground squirrels come in Discussion, and also a comparison between species is mentioned therein, Line 358-371.

3# Line 82: change human to humans

►Done. Line 83. 

Methods

4# Line 123: add “have” in front of “access” in both occurrences

► Done. Line 124. 

5# Line 129: was blood collection done similarly for both hibernating and alert animals? Some anesthetics don’t work as well on hibernating animals due to decreased circulation and metabolism.

► Yes, it was similar for both. Induction of anesthesia to hibernating bears takes slightly longer time than active phase bears. Our protocol has been already established for captive Japanese black bears and is consistent with the protocol of anesthesia for black bears stated in the scientific literature. For example, (Source: Zoo Animal and Wildlife Immobilization and Anesthesia, 2nd edition, Editors: West, G., Heard, D., Caulkett, N. (2014) Blackwell, Page 605, Under topic “American Black Bears”) states that “Medetomidine-zolazepam-tiletamine will induce a rapid onset of immobilization; it can be delivered in a small volume and it is readily reversible…”. 

Also, a similar method has been reported for anesthesia of hibernating bears, Source: Evans et al., 2012, Capture, anesthesia, and disturbance of free-ranging brown bears (Ursus arctos) during hibernation, https://doi.org/10.1371/journal.pone.0040520

6# Line 131: if blood was collected in EDTA this would produce plasma. Was clotting initiated to form serum? Or was plasma actually used in the experiments.

►The sentence reads: “Blood was collected from the jugular vein into plain tubes and EDTA containing tubes for serum collection and peripheral blood mononuclear cells (PBMCs) isolation, respectively.” This means in order to collect serum (to enrich the medium) we used plain tubes. For PBMCs collection, we used EDTA tubes. We used SERUM in the medium. Line 133.

7# Line 174: Change “Next day of” to “The day after” For the experimental design (table 1 and lines 188-194)

►Done. Line 177. 

I have a few concerns with the experimental design.

8# 1. While an n=3 for PBMCs is fairly low an n=1 for ADSC is unacceptably low. It also appears that individual PBMC cell preps were incubated with serum from individual animals. Were any experiments performed where a single cell prep was incubated with each of the different serum samples to ensure there were not differences between serum samples. Or a pooled serum sample could be used in common with all three cell preps. As the experiment is designed it is difficult to tell if any variation is due

to the cell prep or the serum.

► The aim of the present study was to check the efficacy of the bears serum to promote osteoclasts formation in a comparative manner; hibernation phase serum Vs active phase serum. To do this, hibernating phase serum from four individuals (HBSM: C, D, H, I), and active phase serum from four individuals (ABSM: D, E, F, G) were assessed separately. These were conducted for PBMCs from three individuals (A, B, C) separately. 

For example: 

PBMCs of Bear A was cultured with ABSM of Bear D

PBMCs of Bear A was cultured with ABSM of Bear E

PBMCs of Bear A was cultured with ABSM of Bear F 

PBMCs of Bear A was cultured with ABSM of Bear G

PBMCs of Bear A was cultured with HBSM of Bear C

PBMCs of Bear A was cultured with HBSM of Bear D

PBMCs of Bear A was cultured with HBSM of Bear H

PBMCs of Bear A was cultured with HBSM of Bear I

Similar procedure were used for PBMCs of Bears B and C. 

Moreover, to check the influence of HBSM and ABSM on another cell lineage culture, we used stem cells driven from adipose tissue (ADSCs). Accordingly, ADSCs were cultured separately with four types of HBSM, and separately with four types of ABSM. This procedure were carried out twice; first passage and second passage. We did the second passage in order to check the influence of a previous culture (first passage) of the proliferation rate of ADSCs. Interestingly, both passages of 4 HBSM and 4 ABSM (which were separately tested) showed very harmonious results. This experiment shows 16 replications of tests.

One more thing; in order to answer the reviewer’s concern regarding the number of ADSCs (only from one bear), we carried out one more test, but on a limited number of HBSM and ABSM due to limited volume of our serum stock. ADSCs of another bear was cultured for the first passage with one HBSM (D) and one ABSM (E), and then, for the second passage ADSCs were cultured with two HBSM (D, H) and two ABSM (E, F). Interestingly the results were in agreement with the ONE BEAR ADSCs stated in the main manuscript as follows below:

With regards to the use of bear serum for cell culture, our study has examined a passable number of tests on serum (with total 8 types) and cells (three types of PBMCs as the main subject). This is comparable with the sample size in previous studies regarding the effect of bears’ serum for cell culture, as shown in the following box:

Rigano et al., 2016 (https://doi.org/10.1007/s00360-016-1050-9), page 8, 10: Four adult bears’ cells were cultured with serum of only two bears. 

Jansen et al., 2016 (https://doi.org/10.1186/s12983-016-0173-x) Page 6, 13: Cells and serum were collected from four bears. 

Chanon et al., 2018 (https://doi.org/10.1038/s41598-018-23891-5), Page 2 , 3: Bears serum from two groups containing four mixtures of serum was used for culture of two groups of human muscle cells.

Overstreet et al., 2004 (https://doi.org/10.1290/1543-706X(2004)40<4:IOOADA>2.0.CO;2), Page 4: Serum from seven active bears and one hibernating bear was used for humans’ osteoblast culture (not mentioned the number of subjects). 

Due to the limitations of experiments on wild animals, and our institutional regulations regarding the demarcated collection of samples from hibernating animals (which do not have access to food , and are exposed to very low ambient temperature), we designed our tests to the maximum possible variations. 

9# 2. The difference in growth between active (ABSM) and hibernating (HBSM) sera is interesting and suggests that ABSM contains an activator or HBSM contains an inhibitor of PBMC differentiation. A classical experimental method to test this is to either heat inactivate each serum or do a mixing experiment. For example if you heat inactivate ABSM and growth no longer occurs it could contain an activator, while if you heat inactivate HBSM and growth does occur then it contained an inhibitor. Similarly, if HBSM contains an inhibitor, adding it to ABSM could decrease growth. There are multiple ways these experiments may not work (heat stable factors, etc.) but they are fairly simple and broad experiments that would allow the authors to draw a more substantial conclusion about their results.

► In order to check the effect of heat inactivation, we checked the recent scientific literature, and found that Rigano et al., 2016 (https://doi.org/10.1007/s00360-016-1050-9) employed bear serum heat inactivation in their experiment for cell culture. According to that study, page 10: serum was heated for 30 min at 65°C and filtered. Therefore, in order to compare our results with the stated work, we followed the same procedure. When we heated (at 65°C for 30 min) 2 ml of 2 active bears serum and 2 ml of 2 hibernating bears serum, the serum completely coagulated, and became solid; a form which could not be used into medium. We repeated this test, one more time, for four more bear serums, and again found the same results. Afterwards, we checked scientific literature regarding the reason for serum coagulation. We found out that heating of serum to temperatures higher than 58-60°C causes coagulation. This form is not reversible; similar to eggs when eggs are boiled. The coagulated serum cannot be used in the medium. The following figures show the coagulated bear serum when heated at 65°C for 30 min:

The scientific literature confirms our observation in that the 65°C heat causes serum coagulation, in the following box: 

Ballou GA, Boyer PD, Luck JM, Lum FG. The heat coagulation of human serum albumin. Journal of Biological Chemistry. 1944 May 1;153(2):589-605. https://pdfs.semanticscholar.org/3532/15f5895f2b7d6d5dcfdb5459b71d0221e53d.pdf

Page 593, Fig. 2. shows the cloud formation in serum samples subjected to temperatures higher than 60° C.

Page 594, Table.1. shows the relation between time, temperature and coagulation. 

And many other similar statements in the article. 

Tekman S, Öner N. Alterations in protein fractions of heated blood serum. Nature. 1966 Oct 22;212(5060):396-. https://www.nature.com/articles/212396a0

Page 396, The samples which were heated to 50° C and 55° C showed an electrophoretic pattern similar to that of unheated blood serum. However, the sample which was heated to 60° C showed initial alteration in the globulin areas (alpha and beta-globulin fractions came together to form a single fraction). The sample which was heated to 65° C showed decreased albumin mobility and no fractionation of the globulin fractions. And samples that were heated to 65° C had a trace of opalescence.

Also other sources with a similar concept:

Donnelly EB, Delaney RA, Kennedy R. Studies on Slaughter Animal Blood Plasma: II. Heat Lability of Blood Plasma Proteins. Irish Journal of Food Science and Technology. 1978 Jan 1:39-44. https://www.jstor.org/stable/25557950?seq=1#metadata_info_tab_contents

Glass GB. The thermal coagulation point of blood serum: III. Correlation with serum proteins, sedimentation rate and Weltmann reaction. The American journal of medicine. 1950 Jun 1;8(6):759-66. https://www.amjmed.com/article/0002-9343(50)90101-X/fulltext 

Therefore, the method of Rigano et al., 2016 was not reproducible, and it was practically erroneous. 

Basically, heat inactivation is a method to inhibit complement reactions, and remove fibrin. To inactivate serum, it should be heated 55-56° C for about 30 min, Source: Simon J, Müller J, Ghazaryan A, Morsbach S, Mailänder V, Landfester K. Protein denaturation caused by heat inactivation detrimentally affects biomolecular corona formation and cellular uptake. Nanoscale. 2018;10(45):21096-105. https://pubs.rsc.org/lv/content/articlehtml/2018/nr/c8nr07424k

Finally, when we followed method “56° C for 30 min”, we found that the serum samples remained normal (retained their fluidity), and were fine to be used in the culture medium. Thereafter, the heat inactivated serums were filtered with 0.22 um filter, and added into medium.

The heat inactivated ABSM and HBSM were used for PBMCs of Bear A. During the 11 days of culture, cells were poorly adhering to the dish and almost majority of the cells were dead, and no multinucleated osteoclasts were formed. We repeated the test with PBMCs of bear C, and added two control positive (two separate intact ABSM which were unheated). Similar to the previous test (= bear A), PBMCs of bear C which were cultured with heat inactivated HBSM were almost all dead (detached/ floating), and PBMCs which were cultured with heat inactivated ABSM contained floating cells, and there was very low cell density, though relatively more than inactivated HBSM. Positive control (two separate intact ABSM which were unheated) showed full differentiation and formed multi-nucleated/ TRAP positive osteoclasts. 

We conducted one more test, to check the effect of heat inactivated serum on ADSCs. To do this, we used the same heat inactivated ABSM and HBSM to culture ADSCs. In both first and second passage, ADSCs proliferated very well; quite similar to those of intact (unheated) ABSM and HBSM. The following figures show the results:

The difference in results between the effect of heat inactivated serum on PBMCs and ADSCs may be due to the deleted components which are essential for PBMCs culture, but not for ADSCs. A previous study has shown the different effects of serum heat inactivation on some sort of cells, Source: Giard DJ. Routine heat inactivation of serum reduces its capacity to promote cell attachment. In vitro cellular & developmental biology. 1987 Oct 1;23(10):691-7. https://link.springer.com/article/10.1007/BF02620982. Giard (1987) stated that calf serum heat inactivation had negative effects on SV-BHK and CV-1 cells, but not much on fibroblasts. 

Based on the current results, it is not feasible to conclude that the difference in “heat labile” serum factors in HBSM Vs ABSM can promote or demote PBMCs differentiation. More importantly, heat inactivated serum (both ABSM and HBSM) indicated normal and almost equal ADSCs proliferation. Thus, since it is difficult to prove or disprove the effect of heat labile serum ingredients in ABSM or HBSM for PBMCs differentiation, and that there are numerous components that are affect by heating serum (e.g. proteins, peptides, vitamins, antioxidants, microRNAs) which does not specify heat-targeted components, we preferred not to state these results in the manuscript. We just briefly stated the heat inactivation results in discussion Line 340-345. Further work is required to delve into the serum ingredients which encourage/discourage bears PBMCs response. 

10# Results: The images are not of great quality, but if they meet the standards of the journal they should be sufficient. My major concern with the results are addressed above regarding the sample size, especially for ADSC, and the mixing and heat inactivation controls.

►We uploaded the figures with maximum possible size and quality. During conversion of file to PDF, the quality may have decreased. 

11# Discussion: A quick summary of the use of PBMCs in in vitro osteoclastogenesis should be added to the introduction. My first thought when reading the introduction is whether this model was physiologically relevant, i.e. how many osteoclasts arise from PBMCs vs. bone marrow. If this is the standard of the field it would be good to mention this up front. 

► Done. Line 94, 95. In addition, the use of PBMCs for osteoclastogenesis has been mentioned in the first paragraph of discussion. 

12# The mixing and heat inactivation experiments are even more important to perform given the mention on line 296 that hibernating serum inhibits chicken OCs formation. The authors come very close to concluding that HBSM might contain an OC inhibitor in lines 304-313. The experiments described above would allow them to test this hypothesis and make this conclusion directly. The rest of the discussion speculates on possible factors that may be lacking in hibernating serum. I think the authors would be advised to first determine if an activator is lacking or an inhibitor is present before testing individual potential lead compounds.

►The effect of heat inactivation was mentioned above.

The “mixing experiment”, however, is very demanding since it requires multiple sets of pair test of active and hibernating individual serums. As mentioned before, there are four types of active phase serum samples (D,E,F,G) and four types of hibernating phase serum samples (C,D,H,I). To test “mixing effect” one should tests multiple combinations (permutations), e.g. D&C, D&D, D&H, D&I, E&C, E&D, E&H, E&I, and so on. As we mentioned before we have had a demarcated amount of sample for the current experiment. This must be noted that the “individual differences” also matter for such cases. That is, the activating factors of an active bear might be weaker or stronger than inhibiting factors of a hibernating bear, or vice versa. This means that the sample size for assessment of “mixing effect” should comprise the comparison several individuals and consider different ages, body weights, and sexes. However, in order to provide a very basic response to the reviewer, we examined a mixture of one active (10%) and one hibernating bear serum (10%). Moreover, a positive control (one unheated 10% ABSM of the same type used for mixing experiment) and a negative control (one unheated 10% HBSM of the same type used for mixing experiment) were checked. We found that: 

- Positive control: PBMCs completely differentiated to multi-nucleated osteoclasts, and was TRAP positive.

- Negative control: PBMCs failed to differentiate to multi-nucleated osteoclasts, and were TRAP negative.

- Mixing experiment (mixture of the abovementioned positive serum [active phase] and negative serum [hibernation phase]): PBMCs were mostly differentiated to multi-nucleated osteoclasts and were TRAP positive. However, there were some multi-nucleated cells which were TRAP negative. These may indicate a co-dominance for both serum types.

We selected this method to compare 10% active phase serum versus 10% hibernating phase serum. However, this made the total serum ratio of our “mixing experiment” to 20% which is different from the 10% serum content in the main text! which is another setback for interpretation of results of mixing experiment. In our test, we only checked the mixture of one active and one hibernation phase serum. With respect to mixing effects, further studies will be required to check not only the differences in individual serum effects with more sample size, but also to check different ratios of the serum implemented in the culture medium. Since our mixing experiment was restricted in terms of sample size, and the results need further evaluation, we preferred not to mention this experiment in the text of the manuscript.

---

## [Decision Letter · Decision Letter 1]

11 Aug 2020

Hibernating bear serum hinders osteoclastogenesis in-vitro

PONE-D-20-13988R1

Dear Dr. Tsubota,

We’re pleased to inform you that your manuscript has been judged scientifically suitable for publication and will be formally accepted for publication once it meets all outstanding technical requirements.

Kind regards,

Xing-Ming Shi, Ph.D

Academic Editor

PLOS ONE

Additional Editor Comments (optional):

Reviewers' comments:

Reviewer's Responses to Questions

**Comments to the Author**

1. If the authors have adequately addressed your comments raised in a previous round of review and you feel that this manuscript is now acceptable for publication, you may indicate that here to bypass the “Comments to the Author” section, enter your conflict of interest statement in the “Confidential to Editor” section, and submit your "Accept" recommendation.

Reviewer #1: (No Response)

Reviewer #2: All comments have been addressed

2. Is the manuscript technically sound, and do the data support the conclusions?

Reviewer #1: (No Response)

Reviewer #2: Yes

3. Has the statistical analysis been performed appropriately and rigorously? 

Reviewer #1: (No Response)

Reviewer #2: Yes

4. Have the authors made all data underlying the findings in their manuscript fully available?

Reviewer #1: (No Response)

Reviewer #2: Yes

5. Is the manuscript presented in an intelligible fashion and written in standard English?

Reviewer #1: (No Response)

Reviewer #2: Yes

6. Review Comments to the Author

Reviewer #1: (No Response)

Reviewer #2: The authors addressed my concerns and attempted heat inactivation and mixing experiments. 65C is probably too high a temperature for serum inactivation, for tissue culture heat inactivation at 55C for 30 minutes is traditional. For mixing experiments it is important to keep the final concentration of serum the same, so 5% of each to give a total of 10%. This is how assays for clotting factor deficiencies or inhibitors are done. Serum samples could also be pooled, similar to a pooled normal plasma, to reduce the number of experiments that need to be done. For future studies this approach could be used to determine if the authors are looking for an inhibitor or an activator, the first step in determining a mechanism of action.

7. PLOS authors have the option to publish the peer review history of their article (what does this mean?). If published, this will include your full peer review and any attached files.

Reviewer #1: No

Reviewer #2: No

---

## [Editor Report · Acceptance letter]

17 Aug 2020

PONE-D-20-13988R1 

Hibernating bear serum hinders osteoclastogenesis *in-vitro*

Dear Dr. Tsubota:

I'm pleased to inform you that your manuscript has been deemed suitable for publication in PLOS ONE. Congratulations! Your manuscript is now with our production department. 

Kind regards, 

on behalf of

Dr Xing-Ming Shi 

Academic Editor

PLOS ONE